# Amino Acid Profiling of Follicular Fluid in Assisted Reproduction Reveals Important Roles of Several Amino Acids in Patients with Insulin Resistance

**DOI:** 10.3390/ijms241512458

**Published:** 2023-08-05

**Authors:** Csilla Kurdi, Vanessza Lelovics, Dávid Hesszenberger, Anikó Lajtai, Ágnes Lakatos, Róbert Herczeg, Krisztina Gödöny, Péter Mauchart, Ákos Várnagy, Gábor L. Kovács, Tamás Kőszegi

**Affiliations:** 1János Szentágothai Research Center, University of Pécs, Ifjúság u. 20, 7624 Pécs, Hungary; kurdi.csilla@pte.hu (C.K.); kovacs.l.gabor@pte.hu (G.L.K.); 2Department of Laboratory Medicine, Medical School, University of Pécs, Ifjúság u. 13, 7624 Pécs, Hungarylajtai.aniko@pte.hu (A.L.);; 3National Laboratory on Human Reproduction, University of Pécs, 7624 Pécs, Hungary; 4Department of Obstetrics and Gynecology, Medical School, University of Pécs, Édesanyák útja 17, 7624 Pécs, Hungary; 5MTA-PTE Human Reproduction Scientific Research Group, University of Pécs, 7624 Pécs, Hungary

**Keywords:** assisted reproduction technology, follicular fluid, metabolomics, insulin resistance, amino acid profile, predictive value

## Abstract

The global prevalence of insulin resistance (IR) is increasing continuously, influencing metabolic parameters and fertility. The metabolic changes due to IR can alter the molecular composition of plasma and other body fluids. Follicular fluid (FF) is derived mainly from plasma, and it is a critical microenvironment for the developing oocytes. It contains various metabolites and amino acids, and the quality of the oocytes is linked at least partially to amino acid metabolism. Our goal was to quantitatively determine the amino acid (AA) profile of FF in IVF patients and to compare IR and non-insulin resistance (NIR) groups to investigate the AA changes in their FF. Using UHPLC-based methods, we quantified the main 20 amino acids from human FF samples in the IR and NIR groups. Several amino acids (aspartate, glycine, glutamate, and cysteine) differed significantly (*p* < 0.05 or less) between the two groups. The most significant alterations between the IR and NIR groups were related to the glutathione metabolic pathway involving glycine, serine, and threonine. Since insulin resistance alters the amino acid composition of the FF, the oocytes may undergo metabolism-induced changes resulting in poor oocyte quality and less fertility in the insulin resistance groups.

## 1. Introduction

The clinical definition of infertility is a disease of the male or female reproductive system defined by the failure to achieve a pregnancy after 1 year of regular unprotected sexual intercourse. Infertility affects approximately 10–15% of reproductive-age couples worldwide [1]. The primary causes of female infertility are typically related to ovulation disorders, tubal problems, and endometriosis. There are, however, other factors (such as metabolic-disorder-related factors and insulin resistance) that can be responsible for “idiopathic infertility” when the cause of infertility is unknown. By understanding the metabolic risk factors, we can gain more information on idiopathic infertility in women, reducing the proportion of unexplained infertility [2].

As a result of unfavorable lifestyle and dietary changes in modern civilizations, there is an increase in the global prevalence of metabolic syndrome [3]. Metabolic syndrome (MetS) is a term used to describe the simultaneous presence of several cardiovascular risk factors such as insulin resistance, obesity, dyslipidemia, and hypertension [4]. The most widely accepted hypothesis for the pathophysiology of the metabolic syndrome is centered around insulin resistance, which is believed to be partially triggered by excessive fatty acid levels resulting from inappropriate lipolysis [5]. Insulin resistance (IR) is a metabolic disorder with impaired insulin signaling and reduced glucose uptake by the target tissues. It is characterized by obesity, type 2 diabetes and hyperinsulinemia, which is a compensatory response to the target tissue insulin resistance [6]. There is convincing evidence that obesity-associated hyperinsulinemia and insulin resistance have a negative effect on fertility. For instance, the decreased weight in obese women experiencing infertility is linked to an improved frequency of ovulation and increased chances of achieving pregnancy. Even among ovulatory women, the higher body mass index (BMI) is associated with reduced rates of spontaneous pregnancy. The underlying mechanism is believed to involve the adverse impact of elevated insulin levels on ovarian function [7]. The prevalence of obesity is increasing because of the combination of reduced exercise, dietary changes, and high calorie intake [8]. Obesity influences all regulatory systems in the human body and can cause reproductive-system-related problems and infertility [9]. The increase in BMI and obesity can be associated with higher risk of developing reproductive problems such as menstrual irregularities, anovulation, subfertility, miscarriage, and negative pregnancy outcomes [10]. Several studies showed that obese women undergoing in vitro fertilization (IVF) experience a decreased ovarian response to controlled ovarian stimulation [11]. Other studies reported significant impairments in the quality of the oocytes and embryos including lower number of oocytes retrieved [12], lower number of mature oocytes [13], poorer oocyte quality with lower fertilization rates [14], and decreased embryo quality [15]. In addition, there is a causal association between maternal obesity and pregnancy complications, with the risk of pregnancy complications increasing with obesity [7]. Maternal complications during the second and third trimester of pregnancy are often attributed to the metabolic syndrome of obesity. 

As it was mentioned earlier, MetS and obesity may have negative effects on the quality of oocytes. The mature oocytes (metaphase II–MII–oocyte) have many roles in reproduction such as supporting the molecular, cellular, and energetic processes of the early embryo development and providing the genetic content for the new offspring. The ability of an embryo to result in a healthy birth is primarily determined by the integrity of the oocyte. During assisted reproductive techniques the mature oocytes are collected from a group of follicles that have developed due to the administration of external gonadotropins. The competent oocyte is achieved through a delicate balance involving factors such as time, proper oxygenation, and hormonal stimulation as well as sufficient supply of energy and micro-nutrients. Various factors like advanced maternal age, exposure to endocrine disruptors or the presence of reactive oxygen species can disrupt this balance at different levels within the ovarian follicle. Consequently, only a limited number of MII oocytes may be retrieved which may influence the ability to result in a successful reproduction. By increasing the quantity of collected MII oocytes the number of viable embryos also rises leading to an improvement in the success rate of in vitro treatments [16].

Recently, the role of oxidative stress in IR was also recognized as a key factor. Oxidative stress is characterized by an excessive presence of endogenous oxidative species that can damage cells and disrupt signal pathways [17]. Produced primarily in the mitochondria and peroxisomes, reactive oxygen species (ROS) such as superoxide, hydrogen peroxide, and hydroxyl radical ions are the main molecules of oxidative stress. Emerging evidence from recent studies concluded that ROS-induced damage directly contributes to the development and progression of various chronic diseases such as IR and type 2 diabetes [18]. It was also reported that both low and high levels of ROS have negative effects on fertility, embryo quality, and outcome of the pregnancy. These findings are consistent with the concept of a “quiet metabolism”, suggesting that there are specific upper and lower thresholds of metabolic activity within which the embryo remains under optimal conditions [19]. 

Metabolomics, a relatively new subfield within the broader field of “omics” focuses on the analysis of low-molecular-weight metabolites, their presence and concentration in different biological fluids. In recent years, it has been used to explore the underlying biological pathways in different diseases and to identify metabolites that can be used as biomarkers for certain diseases. Metabolomic studies have employed various matrices, including blood, urine, saliva, and more specific biofluids such as follicular fluid [20]. The follicular fluid (FF) is formed by the passage of blood plasma constituents across the blood–follicular barrier. This mechanism is influenced by the secretory actions of granulosa and theca cells [21]. FF serves as a critical microenvironment for the development and maturation of oocytes. It contains essential metabolites such as growth factors, cytokines, energy substrates, amino acids, steroids, and various lipids including cholesterol. These metabolites accumulate within the oocytes and play a vital role in their growth and development [22]. The composition of FF has been found to have an important effect on the oocyte quality and embryo development. The amino acid composition of the FF might be related to the developmental competence of the oocytes [23]. Metabolic alterations seen in the follicular fluid can arise from changes in the plasma metabolites or be influenced by the selective filtering of the granulosa cells. The FF comprises crucial metabolites necessary for oocyte growth and development, serving as an indicator of oocyte quality and embryo viability [21]. Recent studies have reported that IR is strongly related to amino acid metabolism, and it seems that plasma amino acid levels may vary during IR. Due to obesity and IR, there are alterations in the levels of amino acids in the plasma in the early stage of lifestyle-related diseases, but fortunately these alterations can be reversed by interventions that improve insulin sensitivity [3]. 

Since metabolic disorders, especially IR, have great effects on the health of oocytes and fertility, we planned to investigate how the metabolic alterations change the amino acid composition of the follicular fluid. We also surveyed the literature to support our results related to metabolic changes on the quality of oocytes and on fertility.

## 2. Results

### 2.1. Demographic and Clinical Data of Patients

In Table 1, the demographic and clinical features of the patients enrolled to our study are shown. 

### 2.2. Amino Acid (AA) Analysis of the Follicular Fluid Samples

A total of 20 amino acids, the building blocks of the proteins, were measured in all FF samples. The concentration values are shown in Appendix A. Glutamine, alanine, and glycine appeared to be the most abundant AAs in the FF which finding is supported by earlier studies [24] showing the physiological roles of these AAs in oocyte development [25].

#### 2.2.1. Comparison of Amino Acid Content According to Insulin Resistance and Non-Insulin Resistance Groups

The quantitative AA results after statistical comparison were expressed as *p* values and are shown in Table 2. The patients were separated into two groups based on their insulin resistance. One group (IR, n = 11) was defined as patients with insulin resistance and the members of the other group (NIR, n = 36) were, in this regard, apparently healthy. Concentrations of three amino acids were found to be significantly altered in the IR group. These were glycine (*p* < 0.001), cysteine (*p* = 0.037), and aspartate. Aspartate level was significantly (*p* = 0.02) higher in the IR group.

#### 2.2.2. Comparison of Amino Acid Contents Based on Body Mass Index (BMI)

The patients were divided into two groups according to their BMI. BMI is calculated as BMI = kg/m^2^, where kg is the person’s weight (in kg) and m^2^ is the height of the person in meters squared. Between 8.5 and 24.9 BMI, the patients were considered to belong to the normal group (n = 21), while in the overweight group, the BMI score was above 25 (n = 22). All the 11 IR patients were in the overweight group and comparison of the IR/NIR groups showed that the patients in the IR group had significantly higher BMI (*p* < 0.001). The concentration of three amino acids were identified as statistically significant, namely aspartate (*p* = 0.02), glutamate (*p* = 0.008), and glycine (*p* = 0.009). Glycine was present in lower concentrations in the overweight group, and the concentration of aspartate and glutamate was higher.

#### 2.2.3. Comparison of Amino Acid Contents Based on the Age of the Patients

In this comparison, patients were separated into two groups by their age. One group (younger, n = 20) contained patients aged 34 and below, and the other group was the older group (n = 27), in which the age of the patients was 35 or above. There was no significant difference in this comparison.

### 2.3. Multivariate PCA and PLS-DA Analysis

The heatmap (Figure 1) in PCA analysis is a visual representation of the relationship between variables and PCs in a multivariate dataset. In this heatmap, each row represents a variable, and each column stands for a PC. The cells of the heatmap display the strength of the relationship between the variable and a PC. The indication of the relationship is based on color gradient or intensity. Strong positive association between the variable and PC is represented by darker or more intense red or blue colors, and weak or negative associations are represented by lighter or less intense colors. Heatmaps can reveal groups of variables that show similar relationships with PCs. Variables that are close to each other in the heatmap and share similar color patterns are likely to have similar effects on the PCs. Principal component analysis scores were evaluated using PLS-DA analysis for IR and NIR patients. The score plots are illustrated in Figure 2.

The score plot provides valuable insights into the grouping, the patterns, and relationships among samples in a multivariate data analysis. Each follicular fluid sample is represented as a data point and the position of the data point in the plot corresponds to its scores on the principal components. An overlap can be observed between the IR and NIR groups in the score plot. It suggests that there is a similarity between the samples from these groups in terms of the measured variables. The overlap indicates that samples in the IR group may share characteristics that are also present in the NIR group.

### 2.4. Potentially Important Metabolites—Biomarker Analysis

Biomarker analysis aims to identify a metabolite or a set of metabolites capable of classifying conditions or disease with high sensitivity (true positive) and specificity (true negative). The amino acids with significant between-group differences were further evaluated using ROC curve analyses. Of the 20 important amino acids, three (glycine, aspartate, and cysteine) were found to have an AUC above 0.7 (Table 3). Glycine and aspartate had *p* values below 0.05 indicating significant differences between the IR and NIR groups.

When analyzing glycine concentrations in the IR and NIR groups, we performed a receiver operating curve (ROC) analysis to calculate the predictive value of glycine and to determine a diagnostic cutoff value. The results are shown in Figure 3a,b.

### 2.5. Metabolic Pathway Analysis

We analyzed certain metabolic pathways between the IR and NIR groups. Using topological analysis, the cutoff value of the metabolic pathway involvement was set to 0.1 and the pathways with the value above 0.01 were selected as potential key metabolic pathways (Figure 4). A total of 13 metabolic pathways were above this value (Table 4). Based on the significance level (*p* < 0.05), glutathione metabolism (*p* < 0.001) had the lowest value, but four other pathways were identified as target pathways, namely glycine, serine, and threonine metabolism, arginine and proline metabolism, histidine metabolism, and glyoxylate and dicarboxylate metabolism.

## 3. Discussion

In this study, we quantified the 20 main amino acids in the follicular fluid of IVF patients and compared the amino acid profile of patients between IR and NIR groups. Based on the comparison of FF amino acid concentration of patients in the IR and NIR groups, several amino acids were found to be significantly altered. Two amino acids (aspartate and glycine) showed differences both in the IR/NIR and normal/overweight groups. In the follicular fluid samples, the concentration of *aspartate* was higher in the IR group, correlating well with previously reports that the level of aspartate is significantly higher in the plasma of MetS patients [3]. Recently, it was also found that, in obese children, the plasma concentration of aspartate was higher due to the impaired glucose tolerance and the aggravated metabolite metabolism [26]. Another study concluded that the elevated levels of aspartate in the plasma can be a strong predictor for prediabetes [27]. Evidence in the literature suggests that aspartate plays a vital role in the energy metabolism of the oocytes by being converted into oxaloacetate, a key intermediate in the tricarboxylic acid cycle. This metabolic pathway is involved in generating energy for cellular processes [28]. It was observed that there was a significant increase in aspartate levels within the cumulus cells throughout the maturation progress, and it is utilized directly for energy production within the cumulus cells or is transferred to the oocytes for energy production [29].

The other amino acid that was significantly altered between the IR/NIR and normal/overweight groups was *glycine*. The follicular fluid samples of IR patients had lower levels of glycine, and similar results were found in the overweight group. Additionally, this amino acid was found to be the best metabolite candidate for classification of IR from follicular fluid samples. Glycine is a non-essential amino acid that plays a crucial role in various biological processes such as neurotransmitter, controlling epigenetics, reproduction, fertility, and metabolic regulation. Glycine is also a precursor for several important metabolites such as glutathione, porphyrins, purines, hem, and creatine [30]. Glycine is also an important amino acid for fully grown oocytes, and it is transported into the cell by glycine transporter (GLYT1). Glycine and cysteine transport increases at the time of oocyte maturation, which may result in the need for glutathione [31]. Previous studies have concluded that the plasma glycine level is significantly lower in patients with obesity and IR compared to healthy individuals [32]. Our results, based on the measurement of amino acid concentrations in the follicular fluid, showed similar characteristics. In other studies, observations suggested that, in the long term, mild glycine deficiency may facilitate the development of metabolic disorders [33]. Studies based on glycine supplementation reported that adding glycine to the diet increased the insulin response and glucose tolerance, and with a proper dose it was remarkably successful in decreasing other metabolic disorders, many inflammatory diseases, a few types of cancers and obesity [30]. Experiments on animals found that glycine supplementation can also improve embryo quality and implantation rates suggesting its potential role in enhancing fertility [34]. The levels of glycine in the FF have been shown to be a good indicator of post-fertilization development [35]. Glycine also has a significant role in pregnancy, and it should be taken into consideration that the de novo synthesis is inadequate to supply the metabolic demand in late pregnancy [36]. 

*Cysteine* was also found to have a significantly lower concentration in the IR group compared to the NIR group. Cysteine can be obtained from the diet but is also synthetized in the body. It is an important source of sulfur in human metabolism and although it is a non-essential AA, the elderly, children, and individuals with certain types of metabolic diseases need to obtain it from the diet. Cysteine itself is a major extracellular antioxidant, and together with glycine and glutamate, they form the glutathione molecule, which is a vital antioxidant [37]. The availability of cysteine is recognized as a rate-limiting factor in the synthesis of glutathione, and this relationship has been extensively documented in clinical and animal studies. Cysteine, alone or together with glutamate and glycine, raises glutathione levels in the oocytes and cumulus cells, promoting maturation [38]. Cysteine supplementation has been shown to enhance the synthesis and levels of glutathione, thus lowering oxidative stress and insulin resistance [39]. The analysis of IR patients’ plasma revealed that cysteine was found to be lower in concentration [39], and we observed the same in the follicular fluid.

*Glutamate* was the only amino acid that was significantly altered in the overweight group but not in the IR group. The concentration of this amino acid was higher in the overweight group than in the normal group. Glutamate is the most abundant amino acid, and it has a fundamental role in the metabolism of amino acids, and it is also a key molecule in the synthesis of glutathione [40]. It was already reported that the plasma levels of glutamate were higher in obese patients, and the elevated levels of this amino acid were associated with an increased risk of cardiovascular diseases, dyslipidemia, and IR. In the follicular fluid samples, we observed that the levels of glutamate were higher in the IR group, but there was no significant difference between the IR/NIR groups. Another study concluded that the elevated glutamate level in the plasma was associated with higher liver fat content and lower insulin sensitivity, and a higher level of glutamate is related to metabolic dysfunctions [41]. Our findings in the follicular fluid were in good agreement with those in the literature. Overweight patients had a significantly higher concentration of glutamate in their follicular fluid samples. High glutamate levels may be the result of altered metabolic function due to obesity. 

The pathway analysis revealed that the most significant alterations between the IR and NIR groups are the glutathione metabolic pathway and the glycine, serine, and threonine metabolism pathway. Glutathione (GSH) is a vital antioxidant that is produced from three amino acids: cysteine (Cys), glycine (Gly), and glutamate (Glu). It has already been claimed that the stability of reproductive cells and tissues relies on maintaining a balance between the production of free radicals and the presence of scavenging antioxidants [42]. GSH plays a crucial role in the maturation of oocytes, fertilization, and the early development of embryos [43]. It has previously been reported that GSH protects eggs from damage caused by oxidative stress, and therefore, oocytes with higher levels of GSH produce healthier embryos [44]. GSH deficiency has been reported to be related to premature ovarian aging [45]. In another study, it was revealed that GSH has antiaging antioxidant properties, and therefore has an impact on egg health [44]. In our experiments in the IR group, significantly lower levels of glycine and cysteine were reported, and this can give rise to lower glutathione levels. It was concluded earlier that, in metabolic conditions associated with enhanced oxidative stress (such as IR), the availability of glycine can be too low to sustain the optimal rate of GSH synthesis [33]. The glycine-related metabolic pathways can also be affected, as we illustrate in Figure 4. In this study, we found several amino acids that are altered in patients with IR and obesity. These alterations may be the results of the metabolic changes due to the aforementioned disorders. 

## 4. Materials and Methods

### 4.1. Patient Enrollment

This study was conducted between May 2021 and February 2023 at the Department of Obstetrics and Gynecology (FF sampling), and at the National Laboratory on Human Reproduction (analytical studies), University of Pécs, Hungary. Detailed information was given to all patients or their next-of-kin regarding our study protocol, while written consent was obtained from all. Exclusion criteria were patients under 18 years of age, unobtainable or withdrawn consent, and autoimmune diseases or overt diabetes. The study protocol was approved by the Regional Research Ethics Committee of the University of Pécs (no. 5273-2/2012/EHR), conforming to the 7th revision of the Helsinki Declarations (2013). All the 47 patients enrolled in this study received assisted reproductive treatment (ART). The inclusion criteria were either male infertility or female infertility caused by tubal problems. Unsuccessful intrauterine insemination (n = 4; marked as unexplained cause of infertility) was also included in this study. The members of the IR group (n = 11) were diagnosed by endocrinologists based on the homeostatic model assessment of insulin resistance (HOMA IR) formula and subjects were grouped by relying on the data of the medical charts of the hospital informational system.

### 4.2. Stimulation and Collection of Follicular Fluid

We used the GnRh agonist triptorelin in both long and short protocols, and cetrorelix in antagonist protocols. Individual doses of rFSH ranging from 150 to 250 IU per day were used for stimulation, depending on the maturity of the follicles. The starting dose was determined based on BMI and age. A maximum daily dose of 300 IU was given to those with a previously determined low response. We supplemented the stimulation with rLH or hMG individually, according to the patient’s age or response. From the 6th day of the cycle, we monitored the follicular maturity using ultrasound every other day. Gonadotropin was administered individually according to the size of the follicles. When at least two follicles exceeded 17 mm in diameter, an injection of 250 µg (6500 IU) of recombinant human chorionic gonadotropin was given to induce final oocyte maturation. Aspiration was performed 36 h later using an ultrasound-guided transvaginal puncture under routine intravenous sedation. The follicular fluid collection was performed during oocyte retrieval procedure and the samples were centrifuged immediately at 6700 g for 10 min at room temperature to remove the erythrocytes and white blood cells. The supernatant was collected and stored at −80 °C until further analyses.

### 4.3. Sample Processing and Measurement

#### 4.3.1. Reagents

For the amino acid analyses, the following chemicals were used: 3-mercaptopropionic acid ≥99.0% (HPLC grade), orto-phtalaldehyde ≥99% (HPLC grade), 9-fluorenylmethyloxycarbonyl chloride (FMOC chloride) ≥99.0% (HPLC grade both from Merck KGaA, Darmstadt, Germany), acetonitrile, ≥ 99.9% HPLC gradient grade, methanol, ≥99.8% HPLC grade and water HPLC gradient grade, both from Fisher Chemical Pittsburgh, Pennsylvania, United States, 20 mM phosphate buffer of pH 6.2. For the UHPLC elution, the mobile phase was acetonitrile/methanol/water solution: 400 mL acetonitrile, 450 mL methanol, 150 mL water. L-Norvaline (Merck, KgaA, Darmstadt, Germany) was used as an internal standard.

#### 4.3.2. Sample Preparation for the UHPLC Measurement

The quantitative amino acid analyses of the FF samples were performed after precipitation of the proteins, fluorescence derivatization of the amino acids, and UHPLC chromatography (Shimadzu Nexera X2 UHPLC System) using a fluorescence detector (RF-20A XS, both from Shimadzu Europa GmbH Duisburg, Germany) and an internal standard (250 µmol/L L-Norvaline). Then, 300 µL ice-cold acetonitrile solution was added to each 200 µL of follicular fluid sample. The samples were vortexed and centrifuged for 4 min at 6100 g (ScanSpeed Mini, Labogene, Allerod, Denmark). After centrifugation, 600 µL of phosphate buffer was added to 300 µL supernatant and the samples were filtered (Millex^®^ GV 4mm Durapore PVDF 0.22 µm, Merck KGaA, Darmstadt, Germany) and inserted into the autosampler module of the device (SIL-30AC Autosampler) in which the temperature was set to 20 °C. 

#### 4.3.3. Derivatization

For the derivatization of the amino acids, 3-mercaptoproprionic acid (MPA) and orto-phtalaldehyde (OPA) were used. In the case of proline, 9-fluorenylmethyloxycarbonyl chloride (FMOC) was applied. As an internal standard, 250 µmol/L L-Norvaline was utilized. We mixed 7.5 µL of sample, 45 µL of MPA, 22 µL of OPA, and 3 µL of L-Norvaline, and incubated it for 1 min. After the incubation, 10 µL of FMOC reagent was added to the mixture and incubated for 2 min. Finally, 5 µL of derivatized sample was injected into the loop of the injector.

#### 4.3.4. Parameters of the UHPLC Method

Aliquots of 5 μL of the samples were injected into the UHPLC system. A reverse-phase 100 × 3.0 mm Kinetex 2.6 μm EVO C18 100 Å (Phenomenex, Torrance, CA, USA) column was used for the separation. The gradient mobile phase was composed of 20 mmol/L phosphate buffer (A) and 40:45:15 acetonitrile: methanol: water solution (B). The flow rate was 1.3 mL/min, and the column temperature was set to 27 °C. The total running time was 15.1 min. The amino acids (except proline) were detected at 450 nm in an RF-20A xs module. In the case of proline, the detection was achieved at 305 nm. The evaluation was performed using Shimadzu LabSolutions 5.97 SP1 software. Each amino acid was identified by the retention time (RT). The concentration of each amino acid was calculated based on the area under curve of the internal standard. The samples were measured in duplicate and the final concentration was calculated from their average. 

### 4.4. Data Analysis

The statistical analysis was performed using SPSS for Windows (version 28.0.0.0, IBM SPSS Statistics, Armonk, New York, USA). The distribution of data was first checked for the normal distribution, but since most of the amino acids did not follow that pattern, non-parametric statistical tests were chosen. To determine the differences between the groups, the Mann–Whitney U-test was performed. Values of *p*< 0.05 were considered statistically significant. 

To evaluate the amino acid concentration results, the MetaboAnalyst (version 5.0, RRID:SCR_015539, Alberta, Canada) web-based tool was utilized. With this tool, multivariate PCA analysis, biomarker analysis, and metabolic pathway analysis were performed. Based on the potential amino acid candidates in the insulin resistance (IR) and non-insulin resistance (NIR) groups, a heatmap was created for unsupervised clustering. To explore the underlying structure and patterns in the dataset, Principal Component Analysis (PCA) was performed. PCA transforms the original variables into a set of uncorrelated components (principal components—PCs) that capture the maximum variance in the data. Following PCA, PLS-DA was employed to examine the discrimination between the IR and NIR groups. PLS-DA combines the concept of PCA and linear regression to model the relationship between the independent (features) and dependent (class membership) variables. For the biomarker analysis, classical univariate ROC curve analysis was performed to identify potential biomarkers and evaluate their performance. ROC curves were generated, and the AUC was calculated to compute the optimal cutoffs of the different amino acids. An AUC value of over 0.7 was established as discriminative power. For the metabolic pathway analysis, the list of the amino acids measured was added to MetaboAnalyst 5.0. The metabolomic pathway related to the amino acids was found by analyzing the topological characteristics of the pathway. The metabolic pathways related to the IR were obtained and mapped, and the diagram of each metabolic pathway was obtained. The impact threshold was set to 0.10. Any pathway beyond this value was classified as a potential target pathway. 

## 5. Conclusions

Follicular fluid is a complex biological fluid that surrounds the oocytes and is derived mainly from plasma. It is a particularly important microenvironment for the development of the oocytes and the quality of these oocytes is linked to amino acid metabolism. Any alteration in the metabolic processes due to IR will alter the amino acid composition of the follicular fluid as well and the oocytes may undergo metabolism-induced changes, and hence it can result in impaired oocyte quality and decreased fertility. IR can cause several negative effects in patients so it would be useful to improve the state of this disease. The combined dietary changes increased physical exercises and glycine/cysteine supplementation may lead to positive metabolic changes in the serum and alterations in the follicular fluid, as well as have a positive impact on the quality of the oocytes and on fertility.

## Figures and Tables

**Figure 1 ijms-24-12458-f001:**
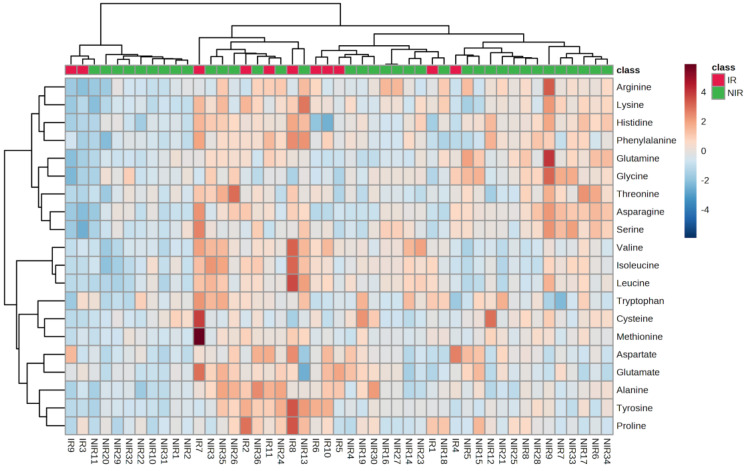
Heatmap of amino acid concentration of IR and NIR samples. Annotations at the bottom of the heatmap show the identification code of the samples.

**Figure 2 ijms-24-12458-f002:**
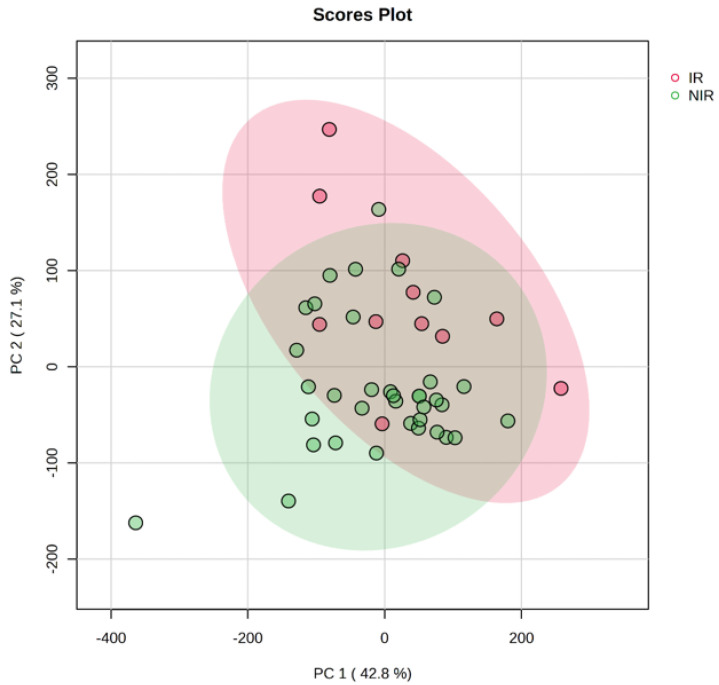
Principal component analysis score plot for IR and NIR patients. The ellipses and shapes show clustering of the samples. The shaded areas indicate the 95% confidence ellipse regions based on the data points for the individual groups.

**Figure 3 ijms-24-12458-f003:**
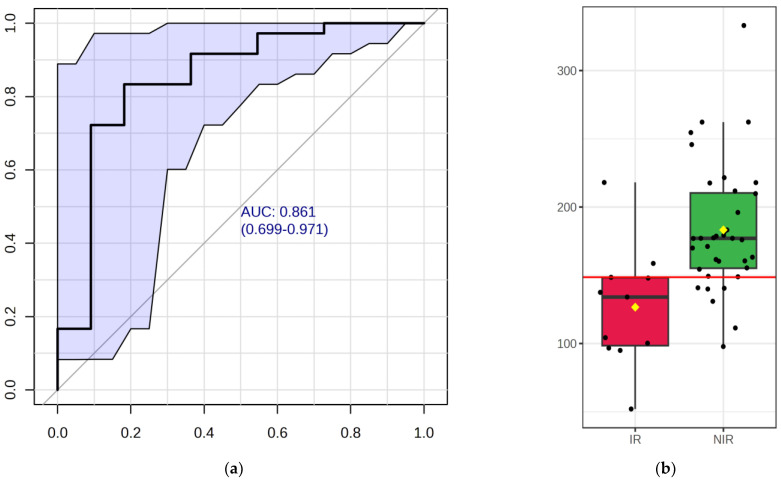
The ROC curve of glycine (**a**). The sensitivity is demonstrated on the *y*-axis and the specificity on the *x*-axis. The area under the curve (AUC) is marked in blue. The boxplot of glycine (**b**) illustrates concentrations of glycine (on the *y*-axis in μmol/L) between IR and NIR groups within the dataset. A horizontal line in red indicates the diagnostic cutoff value. Red box indicates the insulin resistance, while green box the non-insulin resistance groups.

**Figure 4 ijms-24-12458-f004:**
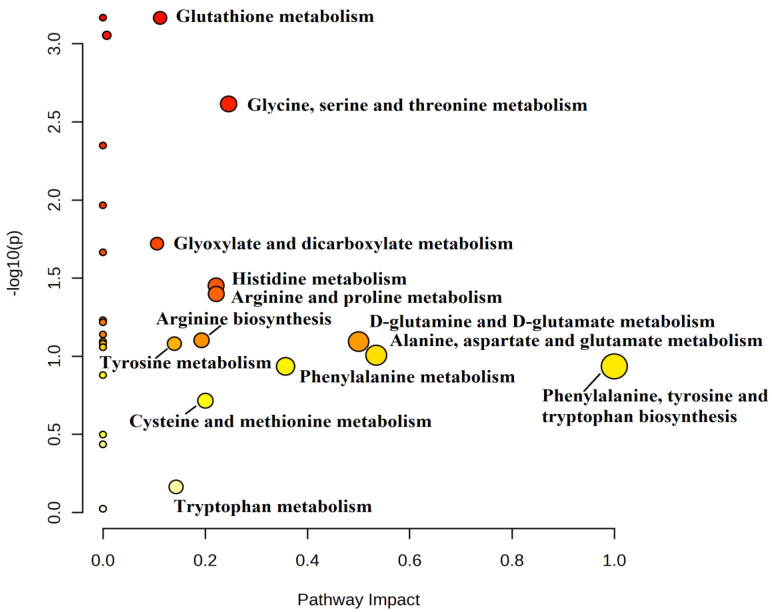
Pathway analysis based on different metabolism of IR and NIR groups. The pathway enrichment analysis is a quantitative analysis using the concentration values of the compounds compared to the list of the compounds used with over-representation analysis. Based on enrichment (*y*-axis) and topology analysis (*x*-axis), this figure illustrates pathways that are significantly changed. The higher impact values represent the relative importance of the pathway, and the size of the circles illustrates the impact of the pathway (the larger circle represents greater pathway enrichment). The color of the circles indicates the significance (the more intense the red color, the lower the *p* value is).

**Table 1 ijms-24-12458-t001:** Clinical characteristics of the patients involved in our research. IR: insulin resistance, NIR: non-insulin resistance.

	IR [Mean ± SD] (*n* = 11)	NIR [Mean ±SD] (*n* = 36)
Age	34.45 ±6.78	35.58 ± 4.69
BMI	32.95 ± 5.55	23.58 ± 4.54
Number of oocytes retrieved	12.27 ± 9.83	10.19 ± 6.81
Number of fertilized oocytes	3.45 ± 3.88	4.42 ± 3.72
Number of IVF cycles	1.9 ± 0.94	2.08 ± 0.77
Cause of infertility		
Male factor	4 (36.36%)	12 (33.3%)
Female factor	3 (27.27%)	15 (42.6%)
Combined male–female	3 (27.27%)	6 (16.6%)
Unexplained	1 (9%)	3 (8.3%)

**Table 2 ijms-24-12458-t002:** Results of the comparison based on different parameters. Each number corresponds to the *p* value, and the significant values are marked in bold. The cut-off was set as *p* < 0.05.

Significance (*p*) Values			
	IR/NIR	BMI	Age
Aspartate	**0.02**	**0.029**	0.297
Glutamate	0.106	**0.008**	0.322
Asparagine	0.42	0.234	0.813
Serine	0.061	0.12	0.182
Glutamine	0.13	0.234	0.813
Histidine	0.72	0.307	0.747
Glycine	**<0.001**	**0.009**	0.228
Threonine	0.951	0.593	0.966
Arginine	0.072	0.264	0.132
Alanine	0.19	0.068	0.312
Tyrosine	0.647	0.274	0.074
Cysteine	**0.037**	0.459	0.54
Valine	0.111	0.068	0.245
Methionine	0.594	0.481	0.813
Tryptophan	0.719	1	0.636
Phenylalanine	0.683	0.576	0.78
Isoleucine	0.351	0.166	0.401
Leucine	0.29	0.174	0.389
Lysine	0.931	0.395	0.254
Proline	0.227	0.369	0.88

**Table 3 ijms-24-12458-t003:** Area under curve (AUC) data obtained from the results of the ROC curves of amino acid analysis of follicular fluid samples.

Amino Acid	AUC	*p* Value
Glycine	0.848	<0.001
Aspartate	0.732	0.004
Cysteine	0.708	0.365
Serine	0.689	0.090
Histidine	0.681	0.029
Arginine	0.681	0.071
Glutamate	0.664	0.058
Valine	0.661	0.060
Glutamine	0.65	0.087
Alanine	0.633	0.316
Proline	0.623	0.046
Leucine	0.608	0.190
Isoleucine	0.595	0.246
Asparagine	0.582	0.425
Methionine	0.555	0.082
Tyrosine	0.547	0.083
Phenylalanine	0.542	0.383
Tryptophan	0.537	0.685
Lysine	0.510	0.945
Threonine	0.507	0.613

**Table 4 ijms-24-12458-t004:** List of the most important pathways with an impact value over 0.1. The impact score refers to a quantitative measure that assesses the significance or importance of a specific amino acid within the metabolic pathway. Based on their *p* value, the significant pathways are marked in bold.

Pathway Name	Impact	*p* Value
Phenylalanine, tyrosine, and tryptophan biosynthesis	1	0.116
Alanine, aspartate, and glutamate metabolism	0.534	0.098
D-Glutamine and D-glutamate metabolism	0.500	0.081
Phenylalanine metabolism	0.357	0.116
Glycine, serine, and threonine metabolism	0.246	**0.002**
Arginine and proline metabolism	0.222	**0.039**
Histidine metabolism	0.221	**0.035**
Cysteine and methionine metabolism	0.200	0.192
Arginine biosynthesis	0.193	0.079
Tryptophan metabolism	0.143	0.685
Tyrosine metabolism	0.140	0.083
Glutathione metabolism	0.112	**0.0006**
Glyoxylate and dicarboxylate metabolism	0.106	**0.019**

## Data Availability

The data that support the findings of this study are available from the corresponding author upon reasonable request.

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
