# Peer review of "Amino Acid Profiling of Follicular Fluid in Assisted Reproduction Reveals Important Roles of Several Amino Acids in Patients with Insulin Resistance"

_ijms, 2023, doi:10.3390/ijms241512458_

Round 1

Reviewer 1 Report

1. The manuscript is well-written and the results are well-presented. However, the English needs to be improved.

2. The inclusion and exclusion criteria should be explicitly stated. Similarly, are the subjects grouped based on medical charts?

There are some typographical errors that need to be corrected prior to publication.

Reviewer 2 Report

The authors have performed an interesting study about the follicular fluid composition of amino acids in IVF.

Overall, the manuscript is well written, and is scientifically and technically sound. My main comment and suggestion for authors is adding some elements about the protocol used in these cases of IVF. 
